# Expression perceptive fields explain individual differences in the recognition of facial emotions
Thomas Murray [1,2] ✉, Nicola Binetti [3,4], Raghav Venkataramaiyer[5], Vinay Namboodiri[5], Darren Cosker[5,6], Essi Viding[7] & Isabelle Mareschal [2]

Humans can use the facial expressions of another to infer their emotional state, although it remains unknown how this process occurs. Here we suppose the presence of perceptive fields within expression space, analogous to feature-tuned receptive-fields of early visual cortex. We developed genetic algorithms to explore a multidimensional space of possible expressions and identify those that individuals associated with different emotions. We next defined perceptive fields as probabilistic maps within expression space, and found that they could predict the emotions that individuals infer from expressions presented in a separate task. We found profound individual variability in their size, location, and specificity, and that individuals with more similar perceptive fields had similar interpretations of the emotion communicated by an expression, providing possible channels for social communication. Modelling perceptive fields therefore provides a predictive framework in which to understand how individuals infer emotions from facial expressions.

Facial expressions can be used to signal a person's emotional state and intentions[1,2], and failure to accurately interpret another's facial expressions impairs social interactions[3], decision making and mental health[4]. However, the long-standing hypothesis that specific facial expressions unambiguously communicate emotional state[5–7], is increasingly challenged by evidence of high variability in how healthy adults interpret facial expressions[8–12]. Attempts to understand the source(s) and impact of this variability have been stymied by tools and methods mainly limited to a small number of stereotyped facial expressions[13], that usually ignore individual differences in the perceptual mechanisms supporting the processing of facial expressions[14].

Recently, researchers have adopted a framework of representing expressions within an expression-space, in which the dimensions correspond to visual properties of the expressions or images[15–19] (see Kriegeskorte, Mur, and Bandettini[20] for a discussion of representational spaces). Such an approach allows for visual variability in the expressions that people may associate with emotions[21], although this is often constrained to the within-category variability found in posed datasets of prototypical expressions. Research has yet to explain how the perceptual encoding of non-prototypical expressions allows for emotion inference. Given the evidence against a universal one-to-one correspondance between expression and emotion[13], any mapping between expressions and emotions must account for (1) the inter-individual variability in the emotions inferred from expressions, and (2) the probabilistic, context-driven nature of emotion inference.

Here, we sought to understand the role of perceptual processes in driving individual differences in facial emotion inference, by exploring expression space with algorithms inspired by evolutionary processes. This type of algorithm allows for efficient exploration within expression-space, and was used to highlight individual differences in the representation of preferred expressions[11]. Participants provided responses to a range of expressions whose positions within expression-space were defined by facial feature weights. This allowed us to model unique probabilistic distributions per participant for different emotion categories, and use these to predict subject-level responses to previously unseen expressions. We refer to these distributions as perceptive fields, analogous to receptive fields in early visual cortex[22]—perceptive fields define regions within expression space in which perception of an emotion is likely. We sought to investigate any individual differences in the size, position, and discriminability of perceptive fields, and investigate how well they can predict emotion inference from unseen facial expressions.

[1]Department of Psychology, University of Cambridge, Cambridge, UK. [2]Department of Psychology, Queen Mary University of London, London, UK. [3]Department of Cognitive Neuroscience, International School for Advanced Studies, Trieste, Italy. [4]Dipartimento di Medicina dei Sistemi, Università degli studi di Roma Tor Vergata, Rome, Italy. [5]Department of Computer Science, University of Bath, Bath, UK. [6]Mixed Reality & AI Lab – Cambridge, Microsoft, Cambridge, UK. [7]Division of Psychology and Language Sciences, University College London, London, UK. ✉e-mail: tom29@cam.ac.uk

## Methods

### Participants

In total 293 participants (mean age = 27.7 y (S.D. = 9.47, range = 18–68); gender = 170 men, 115 women, 1 non-binary, 7 undisclosed) did the genetic algorithm task of which thirty-five participants (mean age = 28.4 y (S.D. = 10.95, range = 18–68); gender = 21 men, 13 women, 1 undisclosed) also completed the categorisation task. Information on gender was provided by participants. Participants were recruited via Prolific (https://www.prolific.co/) to complete the genetic algorithm task, then were recontacted at a later date ( > 6 months) to complete the second task (categorisation task) via Pavlovia (https://pavlovia.org/). All participants provided informed consent and were paid at a rate of £7.50 p/h. Ethical approval was granted by the Queen Mary University Research Ethics Committee (QMERC2019/81) and the University College London Research Ethics Committee (BUCNI-BBK-16-002).

### Genetic algorithm task

Participants 'evolved' facial expressions with a toolkit we developed that uses a Genetic Algorithm to generate facial expressions. We previously used this task to highlight the profound individual differences in people's beliefs about how emotions are expressed by faces[11]. Details of the task procedure can be found in our previous study[11], and details of the algorithm can be found in an archived technical report[23].

**Task procedure**. Before each facial emotion evolution, participants were given a target emotion (anger, fear, happiness, and sadness). On each iteration of the genetic algorithm, participants were presented with 10 same identity facial expressions in a $2 \times 5$ grid. Participants were required to select the expressions that they believed displayed the target emotion, with no constraints on the number of selections allowed, using checkboxes displayed in a separate window. After making this initial selection, participants then chose a single expression from the selected faces that best matched the target emotion using a drop down menu. Participants then pressed a button labelled 'next' to move to the next iteration, where the procedure was repeated. Participants completed between 11 and 15 iterations for each target emotion. The GA evolved the selected expressions on each iteration (or generation) to create new expressions within the set of faces presented in subsequent generation. Apart from the first iteration, the set of 10 facial expressions on subsequent iterations comprised five expressions that had been evolved from selected expressions from the preceding iteration, the single best match from the previous generation, and four new expressions. Importantly, if the participant believed no expressions in the first generation matched the target, they were allowed to restart the session so that a new set of expressions were displayed. These two factors ensured that the participants believed that at least one face on each generation matched the target.

In order to build the perceptive fields that an individual associated with a particular emotion, we used all facial expressions selected by participants for a given target emotion across all generations, applying equal weight to every selected expression. As there were up to 15 iterations and no constraints on the number of expression selections permitted, the potential number of facial expressions available to define the range of expression space per participant could vary from 11 to 150.

**Genetic algorithm**. The genetic algorithm controls the synthesis of facial expressions by loosely encoding the facial action units as blendshapes. Therefore an expression is formally defined as a set of weights over these blendshapes, known as blendshape weights or blendshape space, and the genetic algorithm works by leveraging linear algebra defined over this blendshape space. Based on the participant's selections, the algorithm will cross-breed, mutate and recombine the blendshapes to generate a new set of refined expressions that is then presented to the participant as the next iteration.

Each blendshape weight is between 0 and 1, and the stimuli consisted of 41 core blendshapes, in addition to the blendshapes that control facial symmetry, head position, gaze direction, and corrective blendshapes to enforce realistic expressions. For this study, expressions were symmetrical, and head position and gaze direction were fixed. On all iterations after the first one, the 10 blendshape vectors in each iteration (one vector for each of the 10 expressions which contains the weights of all blendshapes that define the expression) were set by processes of *selection, cross-breeding, mutation, replacement*, and *population diversity boosting*.

*Selection* refers to selection of vectors of blendshape weights (by the participant) to propagate through to the next iteration. The algorithm employs 'elitism', so that propagation of the single best match from each iteration is guaranteed. *Cross-breeding* is carried out through 'uniform crossover' (where blendshapes are independently mutated, rather than clustered within segments), and 'arithmetic recombination' (where the weight of blendshape pairs are averaged). The cross-breeding process occurred for random pairs of parent vectors (i.e. selections from the previous iteration), and the chance of cross-breeding for each blendshape was set at 50%. The algorithm incorporates *mutation*, where the weight of randomly selected blendshapes is drawn from a uniform distribution between 0 and 1. *Replacement* was implemented, such that the parent blendshape vectors were replaced within the subsequent iterations. Finally, the algorithm employed *population diversity boosting* where four new expressions (defined by blendshape vectors) were included within each iteration.

A facial 3D mesh was defined by a set of vertices with Cartesian coordinates and triangular faces. The blendshape vector that defines the expression was applied to the 3D mesh by deforming the coordinates in accordance with a facial blendshape model[24]. The texture applied to the 3D mesh was created by a professional digital artist. We refer the readers to the technical paper[23] for more details of the algorithm and stimulus creation.

### Expression categorisation task

Facial stimuli in the categorisation task comprised 40 examples each of angry, fearful, happy, and sad expressions (160 stimuli in total), that had been created using the genetic algorithm by a separate group of participants. On each trial, participants viewed a single genetic algorithm face and categorised expressions in a forced-choice paradigm, using the four arrow keys to label the expression, with no constraints on the response time. The stimulus presentation order was randomised.

### Statistics and reproducibility

**Modelling perceptive fields**. Multivariate gaussian Kernel Density Estimation (gKDE) was used to model a perceptive field for each of the four expressions, separately for all 293 participants. Gaussian KDE estimates the underlying continuous probability density function (PDF) from discrete data, by smoothing with a gaussian kernel to construct a function from the weighted averages of the data within a sliding window. KDE is non-parametric as it makes no assumptions about the shape of the underlying density function, and can be applied to both univariate and multivariate datasets[25]. Here, we applied KDE to the blendshape data of all the selected faces in the genetic algorithm task for each expression category. We weighted all selections equally when applying the KDE, as this makes no assumption about the fit of any expression to a participant's beliefs, however see Supplementary note 3 for analyses that weights the KDE by the generation in which the expression appeared. Following Scott[26], we reduced the dimensionality of the data before running KDE using principal components analysis (PCA), reducing the 41 dimensions corresponding to the core blendshapes to 10 dimensions. Next, KDE was used on the data along these 10 principal components to estimate a probability density function for each emotion for each participant, using Scott's factor ($n^{(-1/(d+4))}$, where $n$ is the number of data points and $d$ is the number of dimensions) to define the bandwidth of the gaussian kernels in the KDE[26]. As KDE fails to fit more frequently to data in higher numbers of dimensions, we chose 10 dimensions in the PCA as an appropriate

trade-off between % variance accounted for and number of bad fits. At $d > 10$, we were forced to exclude multiple participants due to bad fits.

**Predicting categorisations.** The four PDFs for each participant were used to estimate four probability-densities for each of the faces in the categorisation task. Responses were predicted for each of the categorisation faces using the probability density estimates (i.e. by taking the distribution with the largest of the four estimates). Prediction accuracy was calculated per participant by finding the proportion of correctly predicted responses across all faces in the categorisation task, and separately for each response label used by the participant. Prediction accuracy was assessed with a one-sample t-test, testing the accuracies across participants against chance (25%). Normality was assessed with a Shapiro-Wilk test, non-parametric tests were used if the data was not normally distributed. Bootstrapping was used to estimate 95% Confidence Intervals (CIs) for all effect sizes.

### Statement of preregistration
This analysis was not preregistered.

### Results
Each of 293 individuals used the toolkit[11] to evolve facial expressions on a 3D photorealistic avatar that represented anger, fear, happiness, or sadness. They did this by selecting which of 10 expressions reflected the target emotion category, and repeating this process for 11–15 iterations. A genetic algorithm (GA) used these selections to determine the expressions displayed on the subsequent trial, by adjusting the weight applied to the blendshapes that control the configuration of each expression. These blendshapes are based on biologically plausible muscle movements—like those captured by the Facial Action Coding System (FACS)—to create expressions that are physically realistic.

We pooled all facial expressions selected by all individuals, across emotions and iterations, to provide a joint expression space (Fig. 1a; total 55,241 expressions; average 47.3/emotion/individual, range 15–140). Each blendshape dimension was standardised (z-scored) and PCA was applied to the resultant matrix of z-scored blendshape weights to reduce the 41

blendshape dimensions to 10, which accounted for 33% of the variance—the first two dimensions were dominated by individual differences in the activation of the brows and the lips (Table 1), consistent with previous reports[8,21,27]. The expressions chosen by each individual formed clear clusters—perceptive fields—in this joint space (Fig. 1b).

To capture each individual's perceptive fields, we used multivariate Gaussian Kernel Density Estimation (gKDE) to model the range of expressions that individual would classify as belonging to the same emotion (this allowed us to define a 'spread' measure Fig. 2a, b). We note that we were

**Table 1 | The 10 blendshapes that contribute most to each of the first two components in the PCA**

| PC1 | | | PC2 | | |
|---|---|---|---|---|---|
| Loadings | FACS name | AU | Loadings | FACS name | AU |
| 0.449 | Lip corner puller | 12 | 0.323 | Lip Funneler | 22 |
| 0.360 | Sharp lip puller | 13 | −0.310 | Lip corner depressor | 15 |
| 0.350 | Mouth stretch | 27 | 0.284 | Jaw drop | 26 |
| −0.271 | Brow lowerer | 4 | 0.273 | Lower lip depressor | 16 |
| 0.221 | Dimpler | 14 | 0.271 | Upper lip raiser | 10 |
| −0.221 | Lip corner depressor | 15 | −0.231 | Inner brow raiser | 1 |
| 0.204 | Lip suck | 28 | 0.230 | Lip Funneler | 22 |
| −0.192 | Upper lip raiser | 10 | 0.220 | Outer brow raiser | 2 |
| −0.160 | Chin raiser | 17 | 0.218 | Mouth stretch | 27 |
| −0.159 | Nasolabial deepener | 11 | 0.213 | Upper lid raiser | 5 |

Blendshape weights (loadings), and their corresponding FACS names and Action Units are listed. *PC1* Principal Component 1, *PC2* Principal Component 2, *AU* Action Unit.

**Fig. 1 | Modelling perceptive fields from selections during the GA task. a** Screenshots of iterations during the GA task. In each iteration, individuals selected expressions that represented the target emotion they were tasked with creating. All expressions selected by all individuals were pooled together and transformed using PCA. **b** Perceptive fields for three sample individuals, modelled as probability density functions (PDFs) fit to the selections made during the GA task, using a multivariate gaussian-KDE along the first two of 10 principle components.

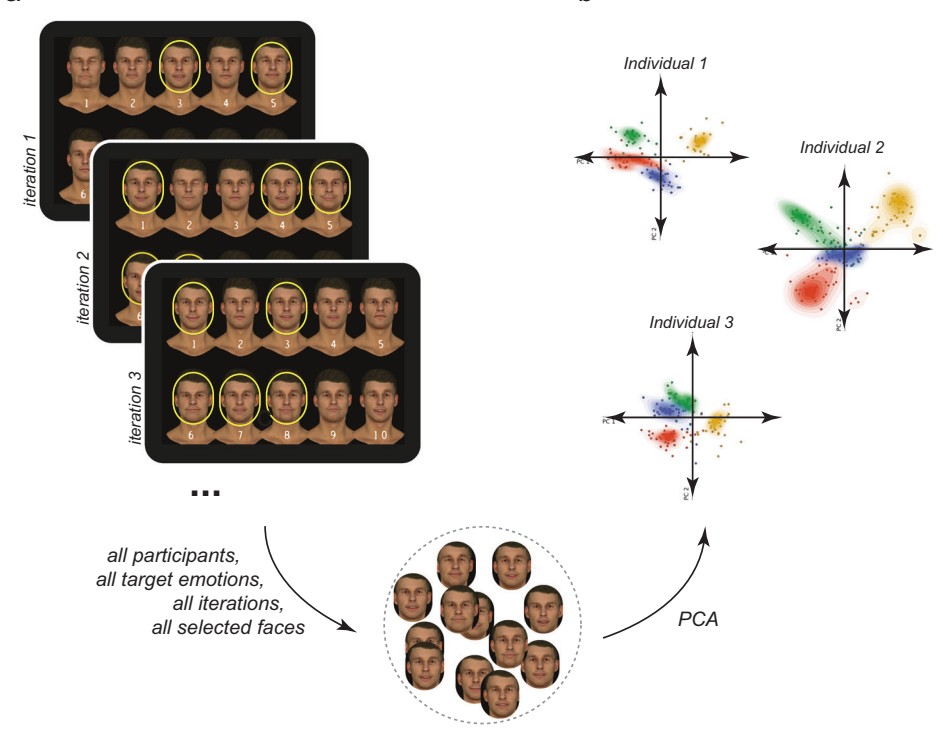

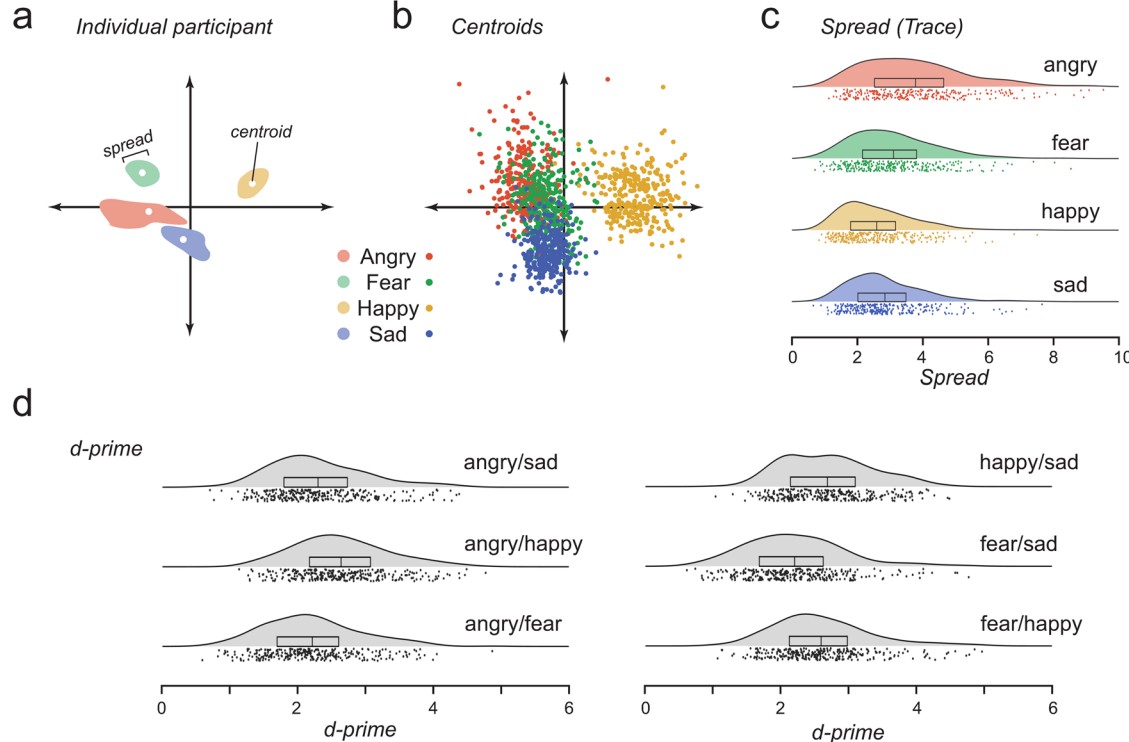

**Fig. 2 | Individual differences in location, spread, and discriminability of perceptive fields. a** Schematic illustration of one individual's perceptive fields represented in the first two dimensions of PCA, for the four emotion categories tested. Each perceptive field is characterised by its centroid and spread. The separation between two perceptive fields and amount of overlap will determine d-prime. **b** Distributions of all participants' perceptive field centroids, separately for each emotion (*n* = 292). Average Euclidean distances between each individual's centroid and all other centroids for each emotion: (Angry: Mean=3.570, S.D. = 0.698; Fear: Mean = 3.407, S.D. = 0.649; Happy: Mean = 2.860, S.D. = 0.565; Sad: Mean = 3.109,

S.D. = 0.617). **c** Distributions of all participants' perceptive field spreads, separately for each emotion (*n* = 292). Horizontal box plots show median and interquartile range. **d** Distributions of d-prime estimates, for each participant, calculated across all combinations of emotion categories (angry-fearful: mean = 2.218, S.D. = 0.710; angry-happy: mean = 2.642, S.D. = 0.696; angry-sad: mean = 2.303, S.D. = 0.708; fear-happy: mean = 2.597, S.D. = 0.706; fear-sad: mean = 2.206, S.D. = 0.727; happy-sad: mean = 2.690, S.D. = 0.663). Horizontal box plots show median and interquartile range (*n* = 292).

unable to fit the gKDE to one participant's selections, so this participant was excluded from the analysis, leaving *N* = 292. Perceptive fields provide a rich description of sensitivity to facial expressions. For example, we found that some individuals have larger perceptive fields, and some have smaller perceptive fields (Fig. 1b), across emotion categories. To characterise spread in the multidimensional expression space, we calculated the sum of the diagonal elements (the trace) of the covariance matrix for each gKDE. On average, perceptive fields for angry expressions were largest (mean=3.782, S.D. = 1.682), followed by fear (Mean = 3.109, S.D. = 1.371), then sad (Mean = 2.846, S.D. = 1.153), then happy (mean = 2.592, S.D. = 1.105). Pairwise comparisons showed significant differences among all pairs (all p's < 0.005; Supplementary Table 1). Linear mixed effects models for each pair of emotion categories (with emotion category and number of expressions selected as fixed effects and subject as a random effect) showed that the effect of emotion category on the spread remained after controlling for differences in the number of faces selected in each category (all p's < 0.003; Supplementary Table 1), suggesting that within-subject differences in the sizes of perceptive fields are not simply driven by differences in the number of selected faces.

We found substantial individual differences in the spread of perceptive fields. The intraclass correlation coefficient in a linear mixed effects model (with spread as the dependent variable, emotion category and number of expressions selected as fixed effects, and subject as random effect) revealed moderate between-subject variability that was not captured by the fixed effects (i.e. variance in the subject-specific baseline measures for spread), capturing 20.7% of the variance in the spread of perceptive fields. Furthermore, spread measures for each emotion were correlated with each other (all p's < 0.05; Supplementary Table 2), suggesting that some

individuals had larger perceptive fields, across emotion categories. This was not simply the result of some individuals selecting more faces than others since a partial correlation factoring in the number of faces selected still revealed significant correlations between all emotion pairs (angry-fear: $\rho(290) = 0.178$, $p = 0.001$, 95%CI [0.076, 0.278]; angry-happy: $\rho(290) = 0.115$, $p = 0.026$, 95%CI [0.001, .224]; angry-sad: $\rho(290) = 0.245$, p < 0.001, 95%CI [0.132, 0.348]; fear-happy: $\rho(290) = 0.224$, $p < 0.001$, 95% CI [0.101, 0.331]; fear-sad: $\rho(290) = 0.255$, $p < 0.001$, 95%CI [0.144, 0.372]; happy-sad: $\rho(290) = 0.193$, $p < 0.001$, 95%CI [0.066, 0.313]; all one-sided). One-sided (positive) correlations were chosen in anticipation of finding individual differences in perceptive field sizes across the emotion categories.

The arrangement of perceptive fields may also explain why some individuals are better at discriminating between the emotions conveyed by facial expressions. We calculated a centroid per perceptive field (as the mean of the selected expressions; Fig. 2a, b) and defined each individual's ability to discriminate emotion categories as the d-prime [d-prime = abs(distance of centroids)/sqrt(mean(spreads))], and estimated it for each individual across every combination of emotion categories (Fig. 2d). We found that on average, people's perceptive fields were better able to discriminate happy from any negative emotion and were worst at discriminating fear from sad. To investigate the distribution of perceptive fields in expression-space, we calculated the average Euclidean Distance (ED) between the centroids of perceptive fields for the same emotion. We found most variability of centroids for angry expressions, followed by fearful, happy, then sad (angry>fearful: W(291) = 16703, $p = 0.001$, $r_{rb} = 0.219$, 95% CI [0.078, 0.334]; angry>happy: W(291) = 4943, $p < 0.001$, $r_{rb} = 0.769$, 95% CI [0.699, 0.853]; angry>sad: W(291) = 9215, $p < 0.001$, $r_{rb} = 0.569$, 95% CI [0.447, 0.658]; fearful>happy: W(291) = 6423, $p < 0.001$, $r_{rb} = 0.700$, 95% CI [0.589, 0.775];

fearful>sad: W(291) = 12154, $p < 0.001$, $r_{rb}$ = 0.432, 95% CI [0.284, 0.521]; sad > happy: W(291) = 13755, $p < 0.001$, $r_{rb}$ = 0.357, 95% CI [0.246, 0.491]). This suggests that perceptive fields for anger may be less consistent across individuals than those for other emotions.

The perceptive fields that we characterise are informative because they predicted how individuals would categorise previously unseen emotional expressions. We presented new expressions to 35 individuals 6–9 months after they had completed the genetic algorithm task and asked them to assign one of the four emotional categories to each of those expressions. We used the individual's gKDE with the largest amplitude at the location of the new expression to predict the emotional category they would choose. We defined prediction accuracy as the proportion of choices that individuals made that matched the gKDE prediction (chance is 25%). Across individuals and all emotion categories, response labels were predicted above chance (mean accuracy = 45.9%, S.D. = 9.70; t(33) = 12.367, $p < 0.001$, d = 2.153, 95% CI [1.660, 3.080] (one-sided)). Accuracy at predicting each emotion label showed that attribution of all labels were predicted above chance (angry: mean accuracy = 44.4%, S.D. = 31.8%, t(33) = 3.500, $p$ = 0.001, d = 0.609, 95% CI [0.317, 0.983]; fearful: mean accuracy=35.8%, S.D. = 26.5, t(33) = 2.338, $p$ = 0.013, d = 0.407, 95% CI [0.091, 0.770]; happy: mean accuracy = 57.1%, S.D. = 28.9, t(33) = 6.375, $p < 0.001$, d = 1.110, 95% CI [0.781, 1.670]; sad: mean accuracy = 42.2%, S.D. = 26.3, t(33) = 3.763, $p < 0.001$, d = 0.655, 95% CI [0.368, 1.060] (all one-sided)).

Knowledge of perceptive fields would be particularly powerful if it also allowed us to predict when individuals will agree about the emotion represented by a facial expression. Agreement between individuals may depend on the overlap of their perceptive fields, so we compared the emotion classifications that an individual made with those predicted by their perceptive fields (e.g. Fig. 3a), and those predicted by the perceptive fields of the other 291 individuals (e.g. Fig. 3b, c). On each of 10,000 iterations we randomly paired individuals, using one individual's perceptive fields to generate the predictions for the other's classifications. On average, another's perceptive fields predicted an individual's classification on 44.4% of iterations, worse than the predictions of the individual's perceptive field (45.9%, $p$ = 0.018), but much better than random (25%, $p < 0.001$). Thus, even in a 10-dimensional space perceptive fields can be unique to the individual, but still show overlap across individuals. We next assessed how well perceptive fields could predict an individual's categorisations in comparison to a population average. By finding the average density for each emotion category across all individual KDEs, the predictions of an 'average KDE' could be generated. Here, we found that the predictions of the group average were weak (26.2%, S.D. = 2.9), but larger than could be expected by chance (t(33) = 2.368, $p$ = 0.012, d = 0.412, 95% CI [0.085, 0.809]), and the

predictions of an individual's categorisations were better than the predictions made by the group average (t(33) = 10.414, $p < 0.001$), further suggesting that perceptive fields are unique to the individual (Supplementary note 4). It should be noted, however, that there is no standard method for averaging across multiple gKDEs, and so the relatively poor performance of the group average may reflect our method of averaging densities across individual KDEs, as the weight of an individual's KDE will be near 0 if their perceptive field does not lie near the stimulus expression.

We next showed that overlap in two people's perceptive fields predicted whether they classified the same test facial expression as displaying the same emotion. For each pair of individuals (n = 35) we defined overlap of perceptive fields for a particular emotion as the integral of the product of the two gKDEs and summed these values across the four emotion categories. We then asked whether this overlap measure could in turn predict the fraction of times that the two individuals agreed on emotion labels across the 160 facial expressions shown; significance was assessed by permuting the overlap and agreement measures 10,000 times. We found that overlap in perceptive fields was correlated with percentage agreement in categorisation of facial expressions ($\rho$(560) = 0.215, p(permuted)<0.001, 95% CI [-0.076, 0.076]), so individuals with more similar perceptive fields will be more likely to agree on the emotional category represented by a particular facial expression.

## Discussion

We showed that the facial expressions that a person judges as belonging to the same emotion category are well described by probability density functions in expression-space, what we call perceptive fields. These perceptive fields are informative and can be used to predict above chance the emotion label an individual assigns to unseen facial expressions during a forced-choice categorisation task. We found profound individual differences in people's perceptive fields: emotion categorisations are better predicted by the locations of an individual's own perceptive fields than another's. Yet, these idiosyncratic perceptive fields also allow social communication: perceptive fields show broad overlap for inter-individual agreement on the emotional category of many expressions, and agreement was better in pairs of individuals with more similar perceptive fields. Our results show that perceptive fields, which can be defined in a low dimensional expression space, provide a predictive framework to understand how people perceive facial expressions.

Previous research has addressed the question of the particular combinations of facial features that contribute to emotion inference[28], using facial feature weights to define a priori models to predict labels that participants might assign to an expression. However, the present study differs in

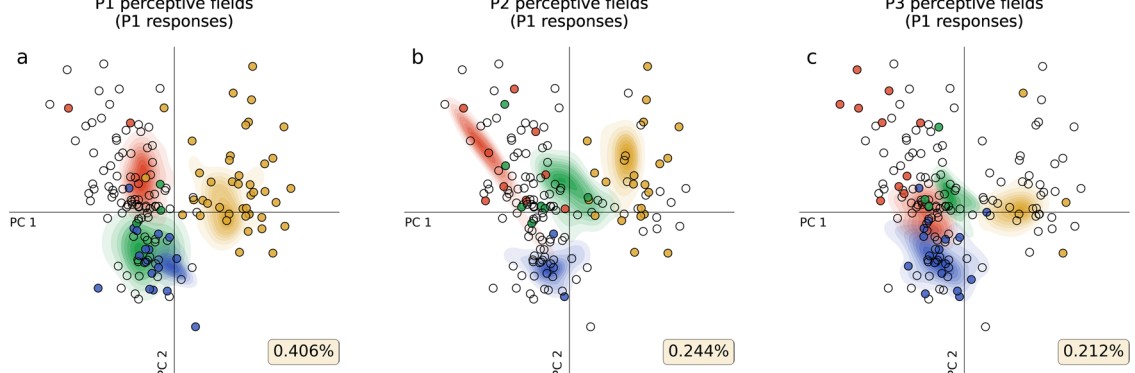

**Fig. 3 | Three examples of predictions within an iteration of the permutation test.** Each panel plots in the two first dimensions in PCA-space, the test facial expressions (circles) superimposed on three different individual's perceptive fields shown as density contours (red = angry, yellow = happy, blue = sad, green = fearful). Filled circles represent expressions where the label given by the sample individual (P1) matched that predicted using the perceptive fields in the same panel. **a** The predictions of P1's labels of expressions, as determined using their own perceptive fields (within-participant predictions). **b, c** The predictions of P1's categorisations of expressions using different individual's (P2 and P3's) perceptive fields (between-participant predictions). Percentages displayed in the bottom right corner of each panel show the percentage of correct predictions.

that it is the first to define these models as subject-specific probability distributions within expression-space, making no assumption about the specific combinations of facial features that a given individual might associate with an emotional expression.

Some commonality in perceptive fields across individuals is expected, because inter-individual agreement in emotion labels is well established, especially within forced choice decisions about facial expressions[14,29]. Importantly, we show that this inter-individual agreement can be explained by similarity in each other's perceptual mechanisms, such that individuals with more similar expression perceptive fields have higher agreement. This framework may therefore provide a tool for quantifying how social communication difficulties may arise between individuals. How people perceive emotions in facial expressions is associated with how they produce those expressions[30–32], although recent evidence suggests that effective social communication may not rely solely on the correspondence between produced expression and internal representation[33,34]. Quantifying perceptive fields may reconcile the role of internal representations of emotion in social communication.

There is intense debate about whether facial expressions reflect distinct, universal emotions (often referred to as the basic emotions theory[6,7]), or whether the perception of facial expressions of emotion is constructed and not unique or universal[35]. Our results align more with the theory of constructed emotion[35] than the basic emotions theory[6,7], as we found that perceptive fields within expression-space are unique to the individual, and that the label that a person assigns to a face depends on where the expression falls within their perceptive fields, rather than the similarity of the expression to a prototype. While we did not explicitly control for the similarity of the expression stimuli to any prototypes, the expressions used within the categorisation task were highly distributed through expression space[11], allowing us to probe responses to expressions that do not necessarily resemble the posed, exaggerated expressions commonly found in stimulus sets.

The difference between the predictive power of perceptive fields within- and between-participants was small, but significant. Constructionist theories of emotional experience and inference highlight the variability in both the expressions that may be produced during an emotional experience, and the emotions that individuals infer from a given expression[13]. Despite this variability, these theories acknowledge that prototypical expressions are reliably labelled across individuals, possibly due to learned associations between expressions and emotions within cultural contexts—analogous to how language shapes colour perception[36,37]. The co-occurring reliability and variability within emotion inference explains our result that both within- and between-subject categorisations were predicted above chance, with an individual's perceptive fields predicting their own categorisations better than those of another. As this difference was relatively small, it suggests that there was a lot of agreement between participants in the labels assigned to expressions.

We found that the attribution of the fear label to facial expressions was not predicted as well as the attribution of the other labels. The distributions of the spreads of the perceptive fields showed that, on average, perceptive fields for angry expressions were larger than those for fear, so one might expect poorer predictive accuracy for anger than fear. However, examining the distribution of pairwise d-prime values suggests that, across participants, the fear-'other' discriminability was poorer than the discriminability of the other category pairs that didn't involve fear. This apparent similarity of the fear perceptive fields to the other perceptive fields may have contributed to the reduced predictive accuracy for the use of the fear label.

An interesting question for future research would be to examine the stability and/or malleability of perceptive fields, and how they may be affected by various state and contextual factors. The emotion that a person might judge a face as expressing can be affected by various state factors such as hormone levels[38,39], mood[40], and state anxiety[41], in addition to external factors such as the context in which the face appears[42–46]. For example, heightened state anxiety is associated with a bias towards labelling expressions as angry[41], so we might expect that people's perceptive fields may shift

within expression-space to accommodate the effects of these state factors. Despite this, in our study, perceptive fields were modelled using data collected during the GA task and were tested on categorisation data that was collected over 6 months later. As such, our results provide evidence for some stability of the unique perceptive fields of individuals.

## Limitations

It is worth noting that the cognitive requirements of the genetic algorithm task differ to those of the labelling (categorisation) task. In the GA task, participants were allowed to restart the GA if they believed no expressions matched the target emotion, and the 'best match' from each generation was always propagated through to the next unaltered. In the categorisation task, participants needed to assign a label to an expression on every trial, regardless of how closely the individual believed the expression displayed the given emotion. So, while labels were assigned to expressions in both types of task, the labelling process differs between the two, as the GA dynamically allows the expression to be adjusted across iterations until it matches the label of the target emotion. Individuals may therefore have rejected facial expressions during the GA task (i.e. if they were not a good match to their own idea of how the emotion is expressed) that, if presented within a forced-choice labelling task, they may have still labelled as displaying that emotion. Similarly, individuals may have viewed expressions in the forced choice labelling task that they judged to display a different expression to the four options, (which would be more likely for those with more narrow perceptive fields), perhaps falling within the perceptive fields of different emotion categories. Our study only probed the representations of four emotion categories, so it would be interesting to examine perceptive fields of other emotions with more subtle distinctions. While only four target emotions were included here, the range of expressions that could be displayed by the GA was not limited to the prototypical expressions typically associated with these emotion categories. Perhaps a more stringent test for the concept of perceptive fields as probability distributions would be to examine how well the perceptive fields can predict ratings of genuineness or intensity along a continuous scale—support for modelling the perceptive fields as distributions in space would be provided if we found that the expressions with higher density were judged as being more genuine.

Finally, our study used static (rather than dynamic) expressions of emotion due to current limitations of genetic algorithms. Dynamic expressions are argued to be more realistic to those encountered in real life[47,48], although others have shown that people can infer emotions from both dynamic and static expressions with comparable efficiency[49]. It would be interesting to consider how perceptive fields emerge or develop over the temporal course of the perception of dynamic expressions.

## Conclusion

Modelling expression perceptive fields offers a predictive framework for emotion inferences from the perception of facial expressions. This framework necessarily accounts for both inter-individual variability, and the probabilistic nature of emotion inferences. We found substantial individual differences in the size, location and specificity of perceptive fields, where some individuals may have consistently large or small perceptive fields across the emotion categories. We also found that the perceptive fields derived with the genetic algorithm were able to predict how people responded in a separate categorisation task. Thus, while perceptual tasks cannot fully disentangle the role of decisional processes from perceptual mechanisms, that genetic algorithms produced perceptive fields capable of predicting responses in the categorisation experiment suggests decision processes are stable across these tasks. Perceptive fields therefore provide a framework within which it is possible to explain and predict the emotions that individuals may infer from facial expressions.

## Data availability

The data to support the results of this study have been reported elsewhere[11]. Each subject's data from the GA and categorisation tasks are available as separate .csv files on the OSF website (https://osf.io/h4q6u/).

## Code availability

The python code (as a notebook file) to replicate the results of the study is available on the OSF website (https://osf.io/h4q6u/). A conda environment file (.yml) is also provided, to use the exact version of the packages used to perform the analyses.

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

## Acknowledgements

This study was funded by a Medical Research Council (MRC) Grant, MR/R01177X/1, awarded to I.M., E.V., and D.C. We thank Samuel Solomon for his insightful contributions to the manuscript.

## Author contributions

T.M.: Conceptualization, Methodology, Formal Analysis, Writing—Original Draft, Visualization; N.B.: Methodology, Writing—Review & Editing; R.V.: Software; V.N.: Software; D.C.: Software, Funding Acquisition; E.V.: Writing—Review & Editing, Funding Acquisition; I.M.: Conceptualization, Methodology, Writing—Review & Editing, Visualization, Funding Acquisition.

## Competing interests

The authors declare no competing interests. The funders had no role in study design, data collection and analysis, decision to publish or preparation of the manuscript.
