## [Peer Review File · Communications Psychology]

19th Jan 24

Dear Dr Murray,

Thank you for your patience during the peer-review process. Your manuscript titled "Expression perceptive fields explain individual differences in the recognition of facial emotions" has now been seen by 3 reviewers, and I include their comments at the end of this message. They find your work of interest, but raised some important points. We are interested in the possibility of publishing your study in Communications Psychology, but would like to consider your responses to these concerns and assess a revised manuscript before we make a final decision on publication.

On occasion we consult with the referees before we finalise our decision. In this case, we further consulted with Reviewer #2 regarding the conceptual and methodological advance of the study in comparison to Snoek et al., (2023), a concern raised by Reviewer #1. This is a summary of their comments:

Reviewer #2:

"I do not agree that Snoek et al., (2023) addressed the same research question. While their analytical framework could potentially be adapted to investigate individual differences, the empirical findings in Snoek et al. do not pertain to individual perceptive fields. Further, if a genetic algorithm can determine individual perceptive fields in a few trials and without having to present 2700 stimuli to a given individual, that could represent a useful contribution. As such, I believe that the approach presented by Murray et al., is novel and could be broadened in future work to consider other basic emotions as well as dynamic stimuli. It seems fair to me to first develop the genetic algorithm approach on static stimuli before doing it with dynamic ones."

We invite you to revise and resubmit your manuscript, along with a point-by-point response to the reviewers. Please highlight all changes in the manuscript text file.

Editorially, we consider:

- Reviewers 2 and 3 each note that there is a force choice aspect to the task (participants have to select at least one face on a trial and there was no "other" option provided). This brings into question whether the perceptive field framework gives a mechanistic account of facial emotion perception (reviewer 2) and might have lead individuals with narrower perceptive fields to include more in their categories than they would otherwise (reviewer 3). Given that individuals make categorical judgments without an intensity scale, the task seems to be a very stringent test of the

prediction that perceptive fields will predict emotion perception performance - clarifying as to how this choice structure may have influenced the results should be considered.

- Details regarding the mutation process of the genetic algorithm (how the set of faces generated in the next generation are determined) and justification for the evolutionary algorithm (e.g., why use faces from every generation rather than the last generation or recent generations to calculate the KDEs) are needed.

- Reviewers 1 and 3 raise the point that the study is limited to 4 basic emotional expressions (anger, fear, happy, sad). It is unclear how/if this approach could be extended to larger (e.g., 6 or more) sets of expressions to demonstrate the broad application of perceptual fields for inference (reviewer 1) and whether the perceptive fields could be malleable based on the task context - for instance if you included a perceptually similar category to fear like surprise would you see the perceptual fields narrow (reviewer 3)?

- Differences in prediction power within and between participants is significant but the effect is very small (less than 2%). What makes this a meaningful effect and why is it so small?

- Are the category differences in perceptive space variability accounted for by differences in the number of faces selected per category.

Please note that your revised manuscript must comply with our formatting and reporting requirements, which are summarized on the following checklist:

Communications Psychology formatting checklist and also in our style and formatting guide
Communications Psychology formatting guide .

Please use the following link to submit your revised manuscript, point-by-point response to the referees' comments (which should be in a separate document to any cover letter) and the completed checklist:

[link redacted]

Please do not hesitate to contact me if you have any questions or would like to discuss these revisions further. We look forward to seeing the revised manuscript and thank you for the opportunity to review your work.

Best regards,

Neil Garrett

Neil Garrett, PhD

Editorial Board Member

Communications Psychology

orcid.org/0000-0003-1440-472X

EDITORIAL POLICIES AND FORMATTING

Editorial Policy: Policy requirements (Download the link to your computer as a PDF.)

* **CODE AVAILABILITY:** All Communications Psychology manuscripts must include a section titled "Code Availability" at the end of the methods section. In the event of publication, we require that the custom analysis code supporting your conclusions is made available in a publicly accessible repository; at publication, we ask you to choose a repository that provides a DOI for the code; the link to the repository and the DOI will need to be included in the Code Availability statement. Publication as Supplementary Information will not suffice. We ask you to prepare code at this stage, to avoid delays later on in the process.

* **DATA AVAILABILITY:**

All Communications Psychology manuscripts must include a section titled "Data Availability" at the end of the Methods section or main text (if no Methods). More information on this policy, is available at <http://www.nature.com/authors/policies/data/data-availability-statements-data-citations.pdf>.

At a minimum the Data availability statement must explain how the data can be obtained and whether there are any restrictions on data sharing. Communications Psychology strongly endorses open sharing of data. If you do make your data openly available, please include in the statement:

We recommend submitting the data to discipline-specific, community-recognized repositories, where possible and a list of recommended repositories is provided at <http://www.nature.com/sdata/policies/repositories>.

If a community resource is unavailable, data can be submitted to generalist repositories such as figshare or Dryad Digital Repository. Please provide a unique identifier for the data (for example a DOI or a permanent URL) in the data availability statement, if possible. If the repository does not provide identifiers, we encourage authors to supply the search terms that will return the data. For data that have been obtained from publicly available sources, please provide a URL and the specific data product name in the data availability statement. Data with a DOI should be further cited in the methods reference section.

REVIEWERS' EXPERTISE:

Reviewer #1

Face recognition

Visual perception

Data Science/Machine Learning

Reviewer #2

Emotional processing/categorisation

Data Science/Machine Learning

Reviewer #3

Emotional processing/categorisation

REVIEWERS' COMMENTS:

Reviewer #1 (Remarks to the Author):

Evaluation: On the positive side, it is true that attempts to understand the source of the variability of how adults across cultures infer facial expressions have been limited by tools and methods that consider only a small number of facial expressions. It is also true that these studies tend to ignore the role of the perceptual mechanisms that process facial expressions. So, studies of the expression space, such as the current ms., are much needed in the field and therefore potentially impactful.

On the less positive side, this ms. faces several prima facie problems that would deserve more careful consideration if a revision was considered. (1) The current study is limited to 4 of the 6 basic expressions and it is unclear how it could be extended to larger sets of expressions; (2) the stimuli are static, limiting their visual realism when the state-of-the-art uses dynamic facial expressions; (3) the derived perceptual fields are not properly validated; (4) others (e.g. Snoek et al., 2023, Science Advances) have already proposed a framework that addresses (1) to (3) above to predict, explain and explore less restricted sets of facial expressions of emotion, questioning the novelty of the present paper, besides its usage of genetic algorithms.

To briefly develop, (1) and (2) do not provide the state-of-the-art realism that the authors advocate. Several methods and published papers can address simple and complex facial expressions well beyond those tested here. To demonstrate the broad application of perceptual fields for inference, the authors should demonstrate applicability to a broader set of dynamic categories of facial expressions. (3) The authors should also successfully validate the perceptual fields with new sets of participants, including possibly using FACS coders for an independent validation of the AUs shown in the resulting blendshapes. (4) Finally, Snoek et al. (2023) have proposed a framework for predicting (and decomposing) the variability of individual participants' judgments of facial expressions that arguably trumps perceptual fields on all of the above dimensions. A comparison of the pros and cons of perceptual fields with the approach developed by Snoek et al. (2023) is therefore warranted.

Reviewer #2 (Remarks to the Author):

Summary

The paper by Murray and colleagues examines whether the recognition of emotions in facial expressions can be modeled as “perceptive fields”, instantiated as gaussian kernel density estimate (gKDE) distributions. The authors assess the differences between perceptive fields for different emotions within and across participants, and test whether gKDE predicts the

categorization of new faces. The authors used an existing dataset where a genetic algorithm was used to evolve “blendshape” parameters to produce sets of facial expressions. They conclude that the perceptive field model successfully explains the categorization of facial expressions of emotion, and that the properties of these fields between and within participants support a constructive theory of emotion.

This is an innovative approach with the potential to have interesting applications in the field. Here are some questions and comments I have.

Major comments

There are no details given for the mutation process of the genetic algorithm. What determines the set of faces generated in the next generation?

The justification for the evolutionary algorithm is unclear. It seems that faces from every generation, rather than the last generation or late generations, were all used to calculate the KDEs. They were not weighted by the fitness of the face in representing an emotion. Can the authors comment on this?

In the conclusion, the authors state that their task differs from a forced-choice paradigm, which would require participants to attribute emotion to faces even if they were not a good representation of their mental prototype of that emotion expression. However, in this experiment, participants were required to select at least one image per generation, thus it also has a forced-choice component (it seems likely that early generations would be less likely to have coherent expressions). Since the selected images from every generation were used to calculate the KDEs, this seems to compromise whether the density estimates accurately represent perceptive fields.

The authors state that the perceptive field framework gives a mechanistic account of facial emotion perception, but they rely on purely statistical modeling and do not reference physical or neural mechanisms. I’m not sure if the account can be said to be mechanistic if it is only looking at probability distributions.

In the discussion, the authors state that their data “align more with the theory of constructed emotion” because they “found that perceptive fields within expression-space are unique to the individual”. I suppose that this claim could be further substantiated by indicating if the mean KDEs can predict significantly worse the judgment of a given individual.

Minor comments

The difference in prediction power within- and between- participants seems very small (less than 2%). The significant p-value does not seem sufficient to conclude that representations are

meaningfully different across people. More could be said about this difference – what makes it meaningful and why is it so small?

The phrase “These perceptive fields are akin to the proposal that facial identity is represented within a multi-dimensional space (face-space, e.g.24), suggesting recognition of facial identity and facial expression is mediated by similar mechanisms” seems somewhat vague and weakly supported. Many things can be modeled within a multidimensional space but have different mechanisms.

The methods state that, for each generation, faces were selected twice: first a set greater than zero, and then a single face from that subset. Why was this selection done twice per generation? Which set was used for what?

It also states that there were no constraints on the number of faces selected, but it seems that participants were required to select at least one, which is a significant constraint as discussed above.

It would be nice to have an analysis that relates prediction accuracy to the amplitude of the KDEs. For example, does the center of the distribution predict new trials better than the edges? This would give a stronger justification for the idea that emotion recognition is implemented as gaussian perceptive fields.

Reviewer #3 (Remarks to the Author):

This manuscript examines whether individuals have varying perceptive fields for how emotion categories are manifest in facial expressions. Not only is there evidence for individual differences, but these predict performance in an emotion perception task months later. This is a very strong contribution to the literature on emotion perception, the analytic approach is innovative and the data highly convincing. Overall, I think this would make an excellent contribution to this journal. I suggest a few potential ways that the manuscript might be further strengthened in a minor revision:

1. The category differences in perceptive space variability are interesting. It would be helpful to know if these differences are accounted for by differences in the number of faces selected per category.
2. The finding that individuals perceive fear in a manner that is not well predicted by the model is interesting. You might expect that anger should have the worst prediction given the strong variability in that category. It would be helpful to see some brief discussion of this finding/tension.

3. I would love to see the authors further characterize the extent of the individual differences observed. The cross-category correlations are helpful, but do not provide a strong intuition about how much variation was observed.

4. The emotion perception task seems to be a very stringent test of the prediction that perceptive fields will predict emotion perception performance, given that individuals are making categorical judgments without an intensity scale. It was helpful to see some acknowledgement of this in the discussion, but I would suggest further discussion to clarify how this choice structure may have influenced the results. (Also, was there no “other” option included?) For example, I can imagine that forcing participants to use the four categories actually led individuals who have more narrow perceptive fields to include more in their categories than they would otherwise.

5. I am also curious to see some discussion of whether the perceptive fields might be malleable based on the task context. For example, is it possible that if you also included a perceptually similar category like surprise (for fear) and disgust (for anger) you would have seen the perceptual fields narrow?

6. It would be helpful to see a bit more explanation on how these data fit better with constructionist accounts. Barrett has written about “population thinking” in emotion (and emotion perception). Perhaps explicitly mentioning this aspect of constructionist theory could help those who are less familiar with the theory understand how your findings are relevant.

A few smaller points that could be clarified by providing more detail:

1. It wasn't clear from the methods section how the PCA was conducted. Figure 1 caption clarifies, but it might be helpful to clarify in the text (i.e., clarify that the PCA was conducted on all participant data). Further, the process for determining that 10 dimensions were optimal was not described.

2. From the description of the task and how d-prime was calculated, I was not expecting a distribution of d-primes for each category pair within each subject. Can this be clarified?

Expression perceptive fields explain individual differences in the recognition of facial emotions – Response to reviewers.

Below we have copied the reviewers' comments (in blue) and explained how we have addressed each, in-line.

All corresponding changes in the manuscript are in a red font.

We note that upon carrying out further analysis in response to the reviewers, we noticed that the PCA was conducted on all expressions *seen* rather than only those *selected* by participants from the 35 participants that completed the categorisation task. We therefore re-ran the analysis on selected faces and from the full sample of participants, and as expected found no significant change in the results (except the prediction of the attribution of the fear label is now significant). We have updated all results in the manuscript accordingly.

In addition to the updated manuscript, we provide a supplementary file, containing results of further analyses that we believe will be of interest to readers.

Reviewer #1:

(1) The current study is limited to 4 of the 6 basic expressions and it is unclear how it could be extended to larger sets of expressions; (2) the stimuli are static, limiting their visual realism when the state-of-the-art uses dynamic facial expressions; (3) the derived perceptual fields are not properly validated; (4) others (e.g. Snoek et al., 2023, Science Advances) have already proposed a framework that addresses (1) to (3) above to predict, explain and explore less restricted sets of facial expressions of emotion, questioning the novelty of the present paper, besides its usage of genetic algorithms.

To briefly develop, (1) and (2) do not provide the state-of-the-art realism that the authors advocate. Several methods and published papers can address simple and complex facial expressions well beyond those tested here. To demonstrate the broad application of perceptual fields for inference, the authors should demonstrate applicability to a broader set of dynamic categories of facial expressions. (3) The authors should also successfully validate the perceptual fields with new sets of participants, including possibly using FACS coders for an independent validation of the AUs shown in the resulting blendshapes. (4) Finally, Snoek et al. (2023) have proposed a framework for predicting (and decomposing) the variability of individual participants' judgments of facial expressions that arguably trumps perceptual fields on all of the above dimensions. A comparison of the pros and cons of perceptual fields with the approach developed by Snoek et al. (2023) is therefore warranted.

We thank the reviewer for their suggestions.

The research by Snoek et al. (2023) is particularly relevant to our study, so we have now included reference to this in our discussion (page 7). Both our research and that of Snoek et al. (2023) share a number of similarities – both studies attempt to define models with combinations of facial features, to predict emotion inference. However, we believe that our research and analytical approach differs to that of Snoek et al in a number of key ways:

1. The focus of our paper was to explore whether subject-level perceptive fields (defined as probability distributions within an expression space) is an appropriate framework for predicting an individual's responses to facial expressions. While both studies examined the combination of facial feature weights that may contribute to emotion inference, Snoek et al. (2023) made no claims about defining regions within an expression space as a framework for predicting emotion inference.
2. The research by Snoek et al. (2023) defined and tested seven *a priori* models to predict emotion inference, based on combinations of facial feature weights – our study did not explicitly define any models beforehand, but rather fit models to the data provided by individuals.
3. While Snoek et al. (2023) make note of individual differences, the focus of their analysis is on defining features that are typically associated with expressions of emotion *across participants*. Individual differences, however, were intrinsic to the motivation for conducting our research.
4. Finally, a key result of ours is the finding that pairs of individuals with more similar perceptive fields agree more on the emotions expressed by a face than pairs of individuals with less similar perceptive fields. This is an important insight into joint facial communication by highlighting an early (perceptual) constraint on how much two people will share the same understanding of an expression. The research by Snoek et al. (2023) made no claims about any overlap between the predictive models of individuals.

We have addressed the four main points raised by the reviewer below:

1. We have added some discussion of our use of four emotion labels to the discussion section (page 8/9), addressing point 1 raised by this reviewer (and point 4 made by reviewer #3, explained in more detail below).
2. We agree that dynamic expressions are more representative of realistically encountered expressions of emotion, and adapting the GA for use with dynamic facial expressions is a planned future development. However the wealth of research using static expressions of emotion should not be disregarded, and given the novelty of our approach, we feel it is warranted to use static faces to contextualise our results in this large field. The reviewer has raised an interesting question about how perceptive fields may temporally emerge, to which we have provided some discussion (page 9).
3. We believe that the re-testing of participants 6 months after the GA task was completed serves as validation for the perceptive fields as models for predicting emotion inference. Additionally, perceptive fields were modelled on the principal components derived from facial feature weights (not the feature weights themselves), and we make no claim about the exact dimensions along which the perceptive fields lie. Therefore, FACS coding the expressions to map blendshapes to action units would not affect our interpretation of the results.
4. We believe that given the fundamental differences between the two studies outlined above, the analytic approach by Snoek et al. (2023) does not “trump perceptual fields”, it addresses different questions. We have, however, provided some discussion comparing our approach to that of Snoek et al. (2023) in the discussion (page 7).

Reviewer #2 (please note that we have added the numbering points to the original comments, for ease of referencing)

Major comments

1. There are no details given for the mutation process of the genetic algorithm. What determines the set of faces generated in the next generation?

We thank the reviewer for the question. While we have attempted to provide a summary of the main features of the algorithm for the readers of this paper, we would stress that this is not a technical paper, and that the details of the algorithm are available elsewhere (Roubtsova et al., 2021). However, we have added some more clarification to the methods to explain the processes used by the GA to perform the cross-breeding of selected expressions (page 10).

2. The justification for the evolutionary algorithm is unclear. It seems that faces from every generation, rather than the last generation or late generations, were all used to calculate the KDEs. They were not weighted by the fitness of the face in representing an emotion. Can the authors comment on this?

We apologise for not providing sufficient justification for our choice of the genetic algorithm. We have now added further justification to the introduction (page 2) and methods section (page 10).

The reviewer has raised an interesting point about weighting the KDE by the fitness of the face. The participant's responses to the expressions were binary (i.e. faces were selected/not-selected), and we did not ask participants for subjective ratings of each expression across iterations, we only obtained this on the last iteration.

One option to address this suggestion would be to weight the KDE by the generation number, as we showed in Binetti et al (2022, PNAS) that expressions converge to the participant's preferred expression as generations increase. We therefore ran the same analysis again, weighting the contribution of each expression to each KDE by the generation that the expression appeared in, and the pattern of results were near-identical. The main difference here is that the prediction of the fear label was not significant at $p < .05$. We believe, however, that treating the data points as binary when modelling the KDE better reflects the selection process, and makes no assumptions about the fit of each face to the participant's representation, so we have not updated the results in the manuscript. These results and discussion may be of interest to some readers, so have included them in the supplementary material. We provide the updated key results using a 'weighted by iteration' approach below:

Predictive accuracy:

All emotions: Mean = 0.454, S.D. = 0.096; $t(33) = 12.201$, $p < .001$
Angry: Mean = 0.454, S.D. = 0.317; $t(33) = 3.707$, $p < .001$
Fear: Mean = 0.326, S.D. = 0.275; $t(33) = 1.584$, $p = .061$
Happy: Mean = 0.575, S.D. = 0.307; $t(33) = 6.072$, $p < .001$
Sad: Mean = 0.408, S.D. = 0.276; $t(33) = 3.291$, $p = 0.001$

Within vs between subject predictions:

Within-subject mean = 45.4%
Between-subject mean = 43.4%
Probability between > within: $p = .001$
Probability between < chance: $p < .001$

Overlap/agreement correlation:

Rho=0.223, p(permutation)<.001

3. In the conclusion, the authors state that their task differs from a forced-choice paradigm, which would require participants to attribute emotion to faces even if they were not a good representation of their mental prototype of that emotion expression. However, in this experiment, participants were required to select at least one image per generation, thus it also has a forced-choice component (it seems likely that early generations would be less likely to have coherent expressions). Since the selected images from every generation were used to calculate the KDEs, this seems to compromise whether the density estimates accurately represent perceptive fields.

We thank the reviewer for raising this point, as we had not been clear enough in our description of the algorithm and task requirements. In our task, participants were allowed to restart the algorithm on the first generation if they believed no expressions matched the target, and the 'best match' from each generation was always propagated through to the next. We had missed this detail from our methods section (so have included it on page 10), and have explained how this means that the labelling processes differ between the two tasks in the discussion (page 8).

4. The authors state that the perceptive field framework gives a mechanistic account of facial emotion perception, but they rely on purely statistical modeling and do not reference physical or neural mechanisms. I'm not sure if the account can be said to be mechanistic if it is only looking at probability distributions.

We agree that the term 'mechanistic' is not appropriate here, and have changed all instances of "mechanistic framework" to "predictive framework" (Page 1 [Abstract], Page 7).

5. In the discussion, the authors state that their data "align more with the theory of constructed emotion" because they "found that perceptive fields within expression-space are unique to the individual". I suppose that this claim could be further substantiated by indicating if the mean KDEs can predict significantly worse the judgment of a given individual.

The reviewer has raised an interesting point, and such a result would support the claim. However, creating new probability density functions as averages of individual functions (in different spaces, with different shapes), poses a difficult challenge. Below we report the results of several attempts at generating predictions of an average KDE.

As the area under the KDE always sums to 1, one approach could be to fit a KDE for each emotion category, over a whole dataset concatenated across the data from individuals. However in testing, the categorisations predicted by these averaged KDEs were no higher than chance:

True accuracy: 45.9%, S.D. = 9.7

Accuracy (averaged KDE): Mean = 25.4%, S.D. = 3.4

True vs average: $t(33) = 10.369$, $p < .001$

Average vs chance: $t(33) = 0.616$, $p = 0.271$

As the KDE was fit across multiple individual datasets (that lie in different locations, and have different sizes), it is not unexpected that this attempt fails. We next tried the same approach, but scaled each individual dataset so that the mean and variance of each set of principal components for each emotion category were equal to the average mean and variance across individuals. Essentially, this approach moved all perceptive fields so that the mean was placed at the average expression chosen across all individuals, and adjusted the variance so that the size was approximately equal across individuals. Again, this approach performed no better than chance:

True accuracy: 45.9%, S.D. = 9.7
Accuracy (averaged KDE): Mean = 25.4, S.D. = 3.7
True vs average: $t(33) = 10.306$, $p < .001$
Average vs chance: $t(33) = 0.649$, $p = .260$

Finally, we tried an approach where predictions of an 'average' were generated by taking the average density across all individual KDEs for each test expression, and using these averaged densities to predict the labels:

True accuracy = 45.9%, S.D. = 9.7
Accuracy (averaged KDE): Mean = 26.2%, S.D. = 2.9
True vs average: $t(33) = 10.414$, $p < .001$
Average vs chance: $t(33) = 2.368$, $p = .012$

Although this approach did not explicitly define an average KDE, the averaged predictions are above chance (but weak), suggesting this approach provides is valid at generating predictions of an 'average perceptive field'. Additionally, the true (within-subject) predictive accuracy was larger than this averaged predictive accuracy, supporting our claim that perceptive fields are unique to the individual.

We believe these analyses may be of interest to readers, so have included it within the supplementary information (and have included reference to this on page 6)

Minor comments

6. The difference in prediction power within- and between- participants seems very small (less than 2%). The significant p-value does not seem sufficient to conclude that representations are meaningfully different across people. More could be said about this difference – what makes it meaningful and why is it so small?

The reviewer has raised an interesting issue here, which we believe is also raised in reviewer #3's comment on constructionist accounts and population thinking (point 6). We have added a paragraph to the discussion (page 8) to address this particular result, explaining it in terms of constructionist accounts of emotion:

"The difference between the predictive power of perceptive fields within- and between- participants was small, but significant. Constructionist theories of emotional experience and inference highlight the variability in both the expressions that may be produced during an emotional experience, and the emotions that individuals infer from a given expression. Despite this variability, these theories acknowledge that prototypical expressions are reliably labelled across individuals, possibly due to learned associations between expressions and emotions within cultural contexts – analogous to how language shapes colour perception.

The co-occurring reliability and variability within emotion inference explains our result that both within- and between-subject categorisations were predicted above chance, with an individual's perceptive fields predicting their own categorisations better than those of another. As this difference was relatively small, it suggests that there was a lot of agreement between participants in the labels assigned to expressions.”

7. The phrase “These perceptive fields are akin to the proposal that facial identity is represented within a multi-dimensional space (face-space, e.g.24), suggesting recognition of facial identity and facial expression is mediated by similar mechanisms” seems somewhat vague and weakly supported. Many things can be modeled within a multidimensional space but have different mechanisms.

We agree that this sentence was weakly supported, and have removed it. We thank the reviewer for the comment, the manuscript reads better without it.

8. The methods state that, for each generation, faces were selected twice: first a set greater than zero, and then a single face from that subset. Why was this selection done twice per generation? Which set was used for what?

This is correct. This was a feature of the GA, as the algorithm required a ‘best match’ from the selected expressions in each generation that propagates through unaltered to the subsequent generation. While our previous work (Binetti et al., 2022, PNAS; Murray et al., 2023, Emotion) used this feature of the algorithm to converge on a single preferred expression of each emotion category (per participant), in this study we made use of all selections made by participants across all generations. We did not account for the ‘best match’ on each generation when calculating the gKDEs, and instead treated all selections equally – we have clarified this on page 10.

9. It also states that there were no constraints on the number of faces selected, but it seems that participants were required to select at least one, which is a significant constraint as discussed above.

The reviewer is correct in that participants must select at least one face per generation, however this may only be an issue in the first iteration (where faces are randomly selected to be in the GA) and they were allowed to restart the algorithm on the first generation if they believed no expressions displayed the target emotion. If they selected at least one face, the one they considered their ‘best match’ was carried through to the next generation. This ensures that the participant was always shown an expression that they had previously selected as displaying the target emotion on subsequent generations. We have added some clarification to the methods (page 10).

10. It would be nice to have an analysis that relates prediction accuracy to the amplitude of the KDEs. For example, does the center of the distribution predict new trials better than the edges? This would give a stronger justification for the idea that emotion recognition is implemented as gaussian perceptive fields.

We agree that an analysis of this type would support the modelling of perceptive fields as probability distributions, however unfortunately, this type of analysis may not be suitable here.

If an expression lies towards the centre of a distribution, it would have a higher density within this distribution than if it fell towards the edges. Each expression also has four densities associated with it – one per distribution. So, if an expression lies towards the centre of one distribution, it may also lie at the edge of other distributions.

The predicted label is defined by the relative density among these four distributions - e.g. if a test expression has a relatively high density within the angry distribution, and relatively low densities within the other distributions, we would predict that the participant would label this test expression as angry. So if the participant labels this face as angry, do we take this as a correct prediction of the angry label, or an incorrect prediction of any other label?

One way of addressing this question would be to ask whether density within perceptive fields could predict responses along a continuous scale, which we have discussed on page 9.

Reviewer #3:

1. The category differences in perceptive space variability are interesting. It would be helpful to know if these differences are accounted for by differences in the number of faces selected per category.

We agree that this would be of interest to the readers. We had not previously included analysis to compare emotion category differences in the spread of the perceptive fields. We have now included this analysis, showing that these differences remain after controlling for differences in the number of selected expressions (page 4).

2. The finding that individuals perceive fear in a manner that is not well predicted by the model is interesting. You might expect that anger should have the worst prediction given the strong variability in that category. It would be helpful to see some brief discussion of this finding/tension.

The reviewer has raised an interesting point, about the relationship between the spread of the perceptive fields, and the accuracy of the predictions. The predictive accuracy of the perceptive fields may depend on the separation/distinction between the perceptive fields for different emotion categories. This is captured by the d-prime calculations – for example, happy faces were predicted most accurately, and happy perceptive fields were most distinct from the others (the happy-other d-primes were the largest amongst the six pairs).

Regarding the poorer prediction of the 'fear' label – the average d-prime for fear-angry and fear-sad were smaller than the average d-prime for angry-sad, suggesting that (on average) the perceptive field for fear overlapped more with the perceptive fields for angry and sad, than angry and sad overlapped with each other. This could suggest that people's representations for fearful expressions are more similar to their representations for angry and sad expressions than angry and sad are to each other, and so the label they assign to a face may be influenced by this.

We have added some discussion of this result to our discussion section (page 8).

3. I would love to see the authors further characterize the extent of the individual differences observed. The cross-category correlations are helpful, but do not provide a strong intuition about how much variation was observed.

Thank you, this is a good suggestion. To address this, we have fit a linear mixed effects model across the whole dataset (with perceptive field spread as the dependent variable, emotion category and number of expressions selected as fixed effects, and subject as random effect). Calculating the intraclass correlation coefficient revealed that 19% of the variance in spread was accounted for by subject-specific differences – we have added this result to the manuscript (page 4), and believe it contributes to the analysis of individual differences in perceptive fields.

4. The emotion perception task seems to be a very stringent test of the prediction that perceptive fields will predict emotion perception performance, given that individuals are making categorical judgments without an intensity scale. It was helpful to see some acknowledgement of this in the discussion, but I would suggest further discussion to clarify how this choice structure may have influenced the results. (Also, was there no “other” option included?) For example, I can imagine that forcing participants to use the four categories actually led individuals who have more narrow perceptive fields to include more in their categories than they would otherwise.

We believe the reviewer has raised an interesting point. It would be an interesting question for future research to address whether the modelling of perceptive fields as probability distributions can predict responses to faces on a continuous scale. We have addressed this in the discussion (page 9): “Perhaps a more stringent test for the concept of perceptive fields as probability distributions would be to examine how well the perceptive fields can predict ratings of genuineness or intensity along a continuous scale – support for modelling the perceptive fields as distributions in space would be provided if we found that the expressions with higher density were judged as being more genuine.”

Regarding the query of adding an option of ‘other’ we decided against this because it is not clear how we would interpret responses in this category, and we worried that participants might resort to using this label whenever they were slightly unsure and inflate this response category.

5. I am also curious to see some discussion of whether the perceptive fields might be malleable based on the task context. For example, is it possible that if you also included a perceptually similar category like surprise (for fear) and disgust (for anger) you would have seen the perceptual fields narrow?

We agree, this is a very interesting point, that is certainly worth exploring. In addition to our discussion of modelling perceptive fields of other emotion categories (page 9), we have added some discussion on the potential ‘malleability’ of perceptive fields, that might reflect various state and contextual factors (page 9). We also note that despite contextual effects on expression perception, we found evidence for some stability/consistency, as participants were recontacted 6 months later to perform the categorisation task.

6. It would be helpful to see a bit more explanation on how these data fit better with constructionist accounts. Barrett has written about “population thinking” in emotion

(and emotion perception). Perhaps explicitly mentioning this aspect of constructionist theory could help those who are less familiar with the theory understand how your findings are relevant.

We thank the reviewer for this suggestion. We believe our result that both within- and between-subject predictions are above chance, and that within-subject predictions slightly outperform between, aligns well with constructionist theories. We have added some additional discussion of these theories (page 8).

A few smaller points that could be clarified by providing more detail:

1. It wasn't clear from the methods section how the PCA was conducted. Figure 1 caption clarifies, but it might be helpful to clarify in the text (i.e., clarify that the PCA was conducted on all participant data). Further, the process for determining that 10 dimensions were optimal was not described.

We have added some clarification to the description of the PCA (page 2).

As the gKDE fails to fit more frequently with higher numbers of dimensions, we chose 10 dimensions as a trade-off between variance accounted for and number of gKDEs that can be fit. At N=10 dimensions, we had to exclude one participant due to bad fits, whereas we had to exclude multiple participants with N>10 dimensions. We have added this justification for choosing 10 dimensions to the methods section (page 11).

2. From the description of the task and how d-prime was calculated, I was not expecting a distribution of d-primers for each category pair within each subject. Can this be clarified?

We thank the reviewer for pointing this out – we calculated a single value for d-prime per category pair, per participant (i.e. 6 per participant), hence leading to distributions of d-prime values. We have clarified how we calculated d-prime (page 4).

In addition to the changes suggested by the reviewers, we have made several other changes to the manuscript.

1. As noted above, we re-ran the PCA and have updated the results accordingly. Specifically:
 - a. We have updated the loadings reported in Table 1 (page 3)
 - b. We have updated our reporting of the size, spread and separation of the perceptive fields, and correlations between these measures (pages 4&5; figure 2)
 - c. We have updated the results reporting the predictive accuracy of perceptive fields, including the permutation tests and within- and between-subject comparisons (page 6)
2. We had originally reported that N=292 in the larger set. This was incorrect, and we apologise for the mistake. The sample size was actually N=293, where one participant was excluded due to a bad fit for the KDE. We have updated the manuscript to explain this.
3. We have added demographic information for both the larger sample of 293 participants and the subset of 35 that completed the categorisation task.

4. We have updated the institutions for two authors (Thomas Murray & Nicola Binetti)
5. We have updated the manuscript to adhere to the Communications Psychology formatting guidelines

13th May 24

Dear Dr Murray,

Your manuscript titled "Expression perceptive fields explain individual differences in the recognition of facial emotions" has now been seen by our reviewers, whose comments appear below. In light of their advice I am delighted to say that we are happy, in principle, to publish a suitably revised version in *Communications Psychology* under the open access CC BY license (Creative Commons Attribution v4.0 International License).

We therefore invite you to revise your paper one last time to address the remaining concerns of our reviewers and a list of editorial requests. At the same time we ask that you edit your manuscript to comply with our format requirements and to maximise the accessibility and therefore the impact of your work.

EDITORIAL REQUESTS:

SUBMISSION INFORMATION:

OPEN ACCESS:

Communications Psychology is a fully open access journal. Articles are made freely accessible on publication under a CC BY license (Creative Commons Attribution 4.0 International License). This

license allows maximum dissemination and re-use of open access materials and is preferred by many research funding bodies.

For further information about article processing charges, open access funding, and advice and support from Nature Research, please visit <https://www.nature.com/commspsychol/article-processing-charges>

At acceptance, you will be provided with instructions for completing this CC BY license on behalf of all authors. This grants us the necessary permissions to publish your paper. Additionally, you will be asked to declare that all required third party permissions have been obtained, and to provide billing information in order to pay the article-processing charge (APC).

* **DATA AVAILABILITY:**

[link redacted]

Best regards,

Jennifer Bellingtier

Jennifer Bellingtier, PhD

Senior Editor

Communications Psychology

Neil Garrett, PhD

Editorial Board Member

Communications Psychology

orcid.org/0000-0003-1440-472X

REVIEWERS' EXPERTISE:

Reviewer #2

Emotional processing/categorisation

Data Science/Machine Learning

Reviewer #3

Emotional processing/categorisation

REVIEWERS' COMMENTS:

Reviewer #2 (Remarks to the Author):

The authors addressed most of my comments. I am although still a bit puzzled by the performance of the mean KDEs (comment 5). In all 3 instances it seems that the mean KDEs are performing poorly, which is a bit in contradiction with the fact that just any individual KDE profile of participants can predict almost as accurately the profile of another person (see comment 6). I would like to see this addressed in the manuscript.

Reviewer #3 (Remarks to the Author):

The authors did an excellent job with this revision and addressed all of my comments/concerns. As I have no new comments, I would recommend publication.